# The Presence of a Single Nuchal Cord in the Third Trimester May Not Affect Tei Index in LGA Fetuses

**DOI:** 10.3390/ijerph20053778

**Published:** 2023-02-21

**Authors:** Julia Murlewska, Przemysław Poszwa, Oskar Sylwestrzak, Maria Respondek-Liberska, Dennis Wood

**Affiliations:** 1Department of Prenatal Cardiology, Polish Mother’s Memorial Hospital Research Institute, 93-338 Lodz, Poland; 2Institute of Materials Technology, Poznan University of Technology, 61-138 Poznan, Poland; 3Department of Diagnoses and Prevention Fetal Malformations, Medical University of Lodz, 90-419 Lodz, Poland; 4Department of Obstetrics and Gynecology, Division of Maternal Fetal Medicine, Sidney Kimmel Medical College at Thomas Jefferson University, Philadelphia, PA 19107, USA

**Keywords:** Tei index, nuchal cord, LGA

## Abstract

(1) Background: The aim of this study was to assess the RV (right ventricle) and LV (left ventricle) Tei index in large for gestational age (LGA) fetuses with a single 360-degree umbilical coil of the umbilical cord around the fetal neck identified by ultrasound in the third trimester of gestation. (2) Methods: The RV and LV Tei index for the cardiac function were measured in 297 singleton pregnancies, and we identified 25 LGA fetuses. There were 48% of LGA fetuses with a nuchal umbilical cord—LGA/NC—larger for gestational age fetuses with a nuchal cord. NC was detected with a color Doppler during a transverse scan of the fetal neck, when the umbilical cord formed a U shape. All fetuses had normal anatomy and normal uterine, placental, umbilical, intracardiac and cerebral Doppler waveforms values for their gestational age. (3) Results: The RV Tei index was significantly higher in the LGA subgroup vs. AGA (0.6 ± 0.2 vs. 0.50 ± 0.2; *p* value = 0.01), but in LGA fetuses with a single coil of the nuchal cord, there were not any significant changes in the Tei indexes. (4) Conclusions: The Tei index might not be impacted by the presence of the nuchal cord in LGA fetuses.

## 1. Background

The nuchal cord occurs in 10 to 29% of fetuses [1]. An umbilical cord around the fetal neck is often diagnosed prenatally by an ultrasound utilizing a color Doppler. A single coil of the nuchal cord is not associated with any resistance to blood flow and alteration of the fetal cardiac performance in the third trimester, as we reported in our previous publication [2]. In this paper, we analyzed the impact of a single coil of an umbilical cord around a fetal neck on the Tei index in those fetuses with LGA and echocardiographically normal heart anatomy and peripheral Doppler waveform parameters.

## 2. Methods

The Tei index for the left and right ventricles was measured prenatally in singleton fetuses ≥ 26 weeks of gestation in patients referred to our center for fetal echocardiography. Chronical maternal diseases (diabetes, cardio-vascular, liver/kidney and any other illnesses) or fetal congenital heart defects, extracardiac malformations or chromosomal malformations were excluded from the study. The Tei index was measured in 297 fetuses by a single sonographer (JM MD, PhD): 158 fetuses were with NC (fetuses with a nuchal cord) and 139 fetuses were with nNC (fetuses without a nuchal cord) as a control group. All studies were performed using GE Voluson E10 bt18 and Voluson E8 bt18, from the General Electric Company equipped with the fetal Echo program and transabdominal RAB6-D and C2-9-D transducer probes. NC was detected with a color Doppler during a transverse scan of the fetal neck, when the umbilical cord formed a “U shape” (Figure 1). Gestational age (GA) was recorded on the basis of the LMP (last menstrual period) and fotal biometry in the 1st trimester, calculated sonographically. Large Gestational Age (LGA) was defined for ≥90th centiles and Small Gestational Age (SGA) was defined for ≤10th centiles. AGA was identified in 263 (88.5%), LGA in 25 (8.41%) and SGA in 9 (3%) fetuses in the study. LGA and SGA were all considered as constitutional growth related to family and maternal conditions. Fetal gestational age, maternal age andthe amniotic fluid index (AFI) were also analyzed. All fetuses had normal peripheral Doppler uterine–placental–cerebral waveforms indices: PI UA—Pulsatility Index of Umbilical Artery, PI DV—Pulsatility Index of Ductus Venosus, PI MCA—Pulsatility Index of Middle Cerebral Artery, PI LUA—Pulsatility Index of Left Uterine Artery and PI RUA—Pulsatility Index of Right Uterine Artery values. They also had normal intracardiac Doppler parameters. At least three Tei index measurement estimations obtaining both of the waveforms on the left and right sides of the fetal heart using the same Doppler gate interrogation were preferred to be acquired in the repeatable values, in the absence of fetal movements and maternal suspended respiration movements, if requested and/or needed. The thermal and mechanical indices of ultrasound machines never exceeded 1, which was consistent with the ALARA rules [3,4]. The Tei index was calculated as (IVCT + IVRT)/ET, where the Tei index is the quotient of or the sum of the duration of the IVCT—isovolumetric contraction time and IVRT—isovolumetric relaxation time, divided by the ejection time (ET) [3,4]. In order to acquire the best repeatable values of Tei indices, we obtained images of the fetal thorax with apical four-chamber views, followed by using valve clicks of the waveforms in the mitral, tricuspid, aortic and pulmonary valve closure artifacts (MV, TV, AoV and PAV clicks). This identifies time periods of the cardiac cycle by maintaining the fastest sweep speed available, lowest possible Doppler gain and high wall motion filter with an angle of insonation close to zero to match the optimal settings with the ultrasound machines. An apical four-chamber view was received and the samples were put at the junctions of the mitral/aortic, tricuspid and pulmonary waveforms [3,4] (Figure 2).

## 3. Statistical Analysis

Bioinformatics using the Python programming language allowed us to develop statistical results (PP). The statistical analysis was prepared with the Python programming language, where the Pandas library was used for data storing and processing along with the calculation of the mean value and standard deviation of the investigated populations. The SciPy library was used to investigate the significance of the differences between populations using the stats module, which includes the possibility of the t-Welch test (t test_ind function). The graphs used in the assessment were prepared using the seaborn library, which allows for the generation of the so-called pair-plots (Figure 3, Figure 4 and Figure 5: see at X-axis and Y-axis). The graphs allow one to examine pairs of individual traits within a given population (diagonally—scatter plots) and to assess the distribution of a given trait in the population (diagonally—histograms). The descriptions of the individual X and Y axes for the scatter plots are given in the most external charts on the left and at the bottom (Figure 3, Figure 4 and Figure 5). This arrangement of the X and Y axes makes the graphs below the histogram and above the histogram represent the same pairs of traits, with the difference that the X and Y axes are swapped. For this reason, the graphs below the diagonal are accompanied by a linear regression diagram with the measurement uncertainty indicated to assess whether a trait on the Y axis in the population is increasing, constant or decreasing, as the trait on the X axis increases [5,6] (Figure 3, Figure 4 and Figure 5).

## 4. Results

Results are shown in Table 1 and Figure 3, Figure 4 and Figure 5. The mean maternal age in our study was: 30 ± 5.4 (mean ± SD—standard deviation), measured in years, and the gestational age was 31 ± 3 (mean ± SD), measured in weeks. The gestational age in the two groups (NC vs. nNC) was similar: (31 ± 3.3 vs. 31 ± 2.8; *p* value = 0.09); the maternal age was also similar (30 ± 5.3 vs. 30 ± 5.5; *p* value = 0.9); the AFI in both groups was similar (17 ± 4 vs. 16.1 ± 4.2; *p* value = 0.4); the Tei index for RV and LV were similar (RV 0.5 ± 0.2 vs. 0.5 ± 0. 2; *p* = 0.8), (LV 0.5 ± 0.1 vs. 0.5 ± 0.2; *p* = 0.4) (Table 1, Figure 3). Within the study group, we have identified subgroups: 263 AGA fetuses, 25 LGA fetuses and 9 SGA fetuses. An umbilical cord around the fetal neck occurred in 45% in AGA fetuses (NC in AGA *n* = 118), in 48% in LGA fetuses (NC in LGA *n* = 14) and 100% in SGA fetuses (NC in SGA *n* = 9). The SGA fetuses group, due to low numbers (*n* = 9), was eliminated from the next part of our analysis. In LGA fetuses vs. AGA fetuses, the RV Tei index was significantly higher (0.6 ± 0.2 vs. 0.5 ± 0.2); *p* = 0.01. The AFI index in the LGA group was significantly higher with respect to the AGA group: (19 ± 2.8 vs. 16 ± 4.1); *p* value = 0.0001. The Tei LV LGA vs. Tei LV AGA (0.6 ± 0.1 vs. 0.5 ± 0.2) were higher, although there was no statistic difference between the groups (*p* = 0.23) (Table 1, Figure 4). In the group of fetuses with LGA, there were 11 with NC and 14 with nNC. The Tei LV was not significantly higher for LGA fetuses with umbilical cord collision (LGA with NC); *p* = 0.009 (LGA + nNC vs. LGA + NC; 0.5 ± 0.1 vs. 0.6 ± 1; *p* value = 0.009), (Table 1, Figure 5).

## 5. Discussion

A nuchal umbilical cord (NC) was observed in our study in 45% of AGA fetuses, 48% of LGA fetuses and 100% of SGA fetuses and was diagnosed by ultrasound, with the addition of a color Doppler, which showed the umbilical cord wrapped around the fetal neck in a “U” nuchal cord configuration. There is only one study, we believe, by Shi from 2017 [7], which mentioned the Tei index of the right ventricle in the group of fetuses under the condition of a nuchal cord. His study group was much smaller comparing with our data (*n* = 55), the gestational weeks were similar (30.2 ± 1.3) and the maternal age was similar: (29.5 ± 2.1); the RV Tei idexes were 0.42 ± 0.04 vs. 0.38 ± 0.05 in the studied and control group (*p* < 0.05); thus, they recorded a significant relationship associated with the effect of the umbilical cord surrounding the fetal neck on the elevation of the Tei index for the right ventricle [7].

In our research data, the RV Tei index was observed as being even higher, but without significant differences depending on the occurrence of NC (NC vs. nNC: 0.5 ± 0.2 vs. 0.5 ± 0.2), (*p* = 0.8), in relation to the work of Shi W. et al. [7]. The differences in Tei indexes in the Shi group and in our study group could be explained by the presence of a subgroup of patients with LGA and a small group of SGA in our study, as well as biometric differences associated with European and Asian fetal populations.

In our previous publication, we also did not notice a significant impact of the nuchal cord on fetal heart function expressed as the Tei index, at the time of the fetal heart examination (at mean gestational age 29 + 4 weeks) and obtaining very similar Tei values for the left and right ventricles, as in this actual analysis [2]. An additional interesting observation in our paper was the statement of the occurrence of the elevated PS MCA in fetuses with a nuchal cord [2], which Sherer DM et al. did not confirm, however [8]. In our paper, the fetal cerebral vascular resistance was not affected by the presence of nuchal cord(s) in the third trimester of pregnancy; however, they examined the mean UA systolic/diastolic ratio (S/D), UA resistance index (RI) and mean fetal MCA S/D and RI, but did not examine the PS MCA [8].

In our current publication, we decided to check how the NC affects the fetal heart function in LGA fetuses; we found a significantly higher RV Tei index and AFI versus AGA fetuses, but not a higher Tei index in LGA fetuses with NC versus LGA fetuses with nNC. We did not analyze RV/LV Tei in SGA fetuses due to the low number of fetuses in this subgroup (*n* = 9). Our finding of 100% of the “umbilical cord collision” in this subgroup of SGA fetuses is reported for the first time and therefore may also be interesting. Our publication paves the way for further research on the analysis of functional parameters of the heart of the fetus with NC depending on the occurrence of LGA or SGA. This needs to be studied in a larger population; although, in the absence of fetal heart defects, the increased Tei index in fetuses with NC and growth impairment appears to be a temporary/transient effect and does not significantly affect the fetuses’ circulatory performance at the time of this study, as well as their postnatal circulatory performance.

## 6. Conclusions

A single coil of umbilical cord around the fetal neck in constitutional LGA fetuses may not be a cause of systolic and diastolic dysfunction of the fetal heart, assessed by the Tei index in this subgroup; however, fetuses with LGA may present higher Tei indices in general.

Therefore, we can assume that NC probably does not significantly affect the increase in Tei index in the group of fetuses with excessive growth in the third trimester of pregnancy, but we need further study results on a larger group of patients to confirm these findings.

## Figures and Tables

**Figure 1 ijerph-20-03778-f001:**
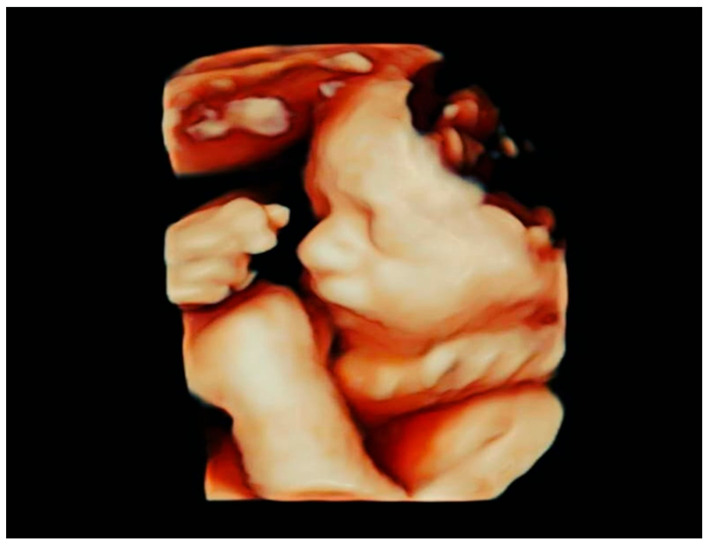
Three-dimensional image showing a nuchal cord.

**Figure 2 ijerph-20-03778-f002:**
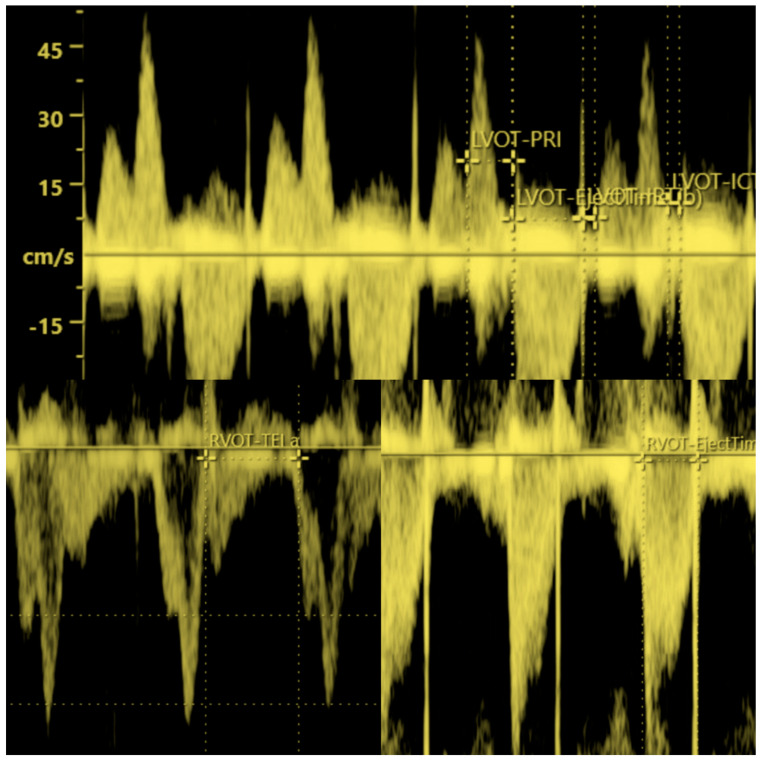
Picture above: Color Doppler waveforms across the mitral and aortic fetal valves (the clicks of aperture and closure of the valves are used to identify time periods of the cardiac cycle. Myocardial Performance Index (Tei index) is defined as isovolumetric contraction time (ICT) plus isovolumetric relaxation time (IRT) divided by the ejection time (ET). The E wave represents passive ventricular filling and A wave represents active ventricular filling. Mechanical PR interval is measured between the onset of atrial contraction (A-wave) and the onset of ventricular contraction (V-wave) and establishes atrioventricular time interval). Picture on the bottom left: Doppler waveforms across the tricuspid fetal valve (pulmonic flow is not evident across the tricuspid valve). E corresponds to early diastolic peak velocity and A wave corresponds to atrial contraction peak velocity. The Doppler clicks of aperture and closure of the valves are used to identify Tei a for the right ventricle. Picture on the bottom right: Measurement of the ejection time of the right fetal ventricle. This photo was taken from the archives of Department of Prenatal Cardiology, Polish Mother’s Memorial Hospital Research Institute, Department of Diagnoses and Prevention Fetal Malformations Medical University of Lodz, Lodz, Poland.

**Figure 3 ijerph-20-03778-f003:**
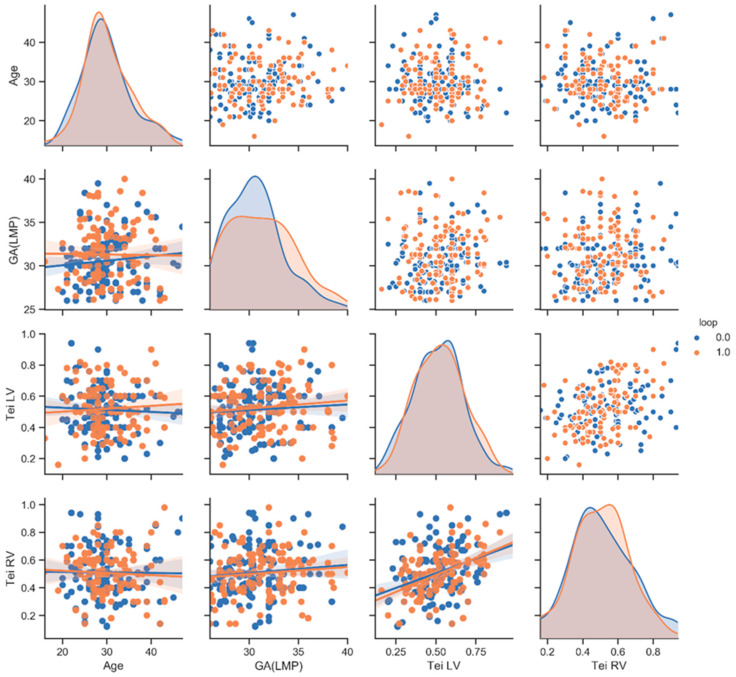
Correlogram of Tei LV, Tei RV, maternal age and gestational age in NC group presented in red dots (study group) (loop = 1.0) (*n* = 139) and in nNC group presented in blue dots (control group) (loop = 0.0) (*n* = 158) visualized with pair-plots in Python. A pair-plot allows one to see both distributions of single variables and relationship between two variables. For more explanation, see statistical analysisin this article, with red dots [5,6].

**Figure 4 ijerph-20-03778-f004:**
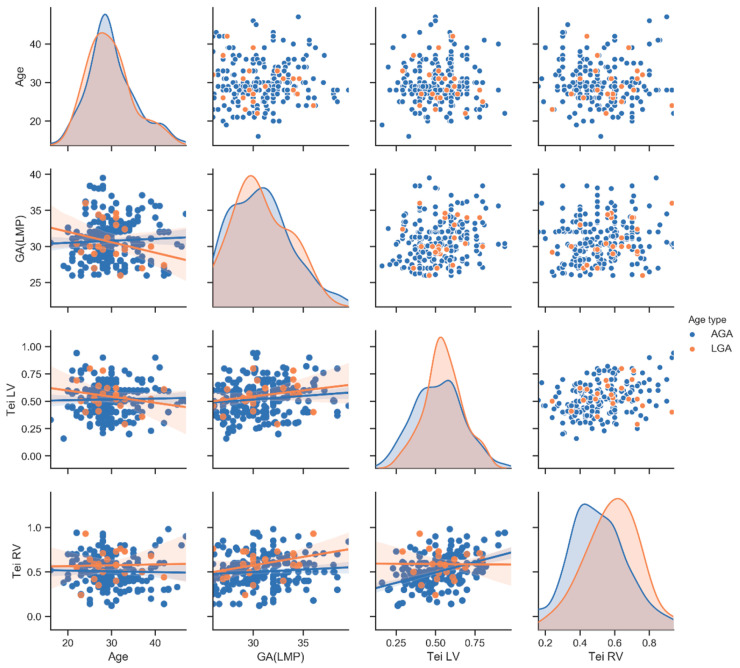
Correlogram of Tei LV, Tei RV, maternal age and gestational age in LGA presented in red dots (*n* = 25) and AGA presented in blue dots (*n* = 263), studied fetuses visualized with pair-plots in Python. A pair-plot allows one to observe both distribution of single variables and relationship between two variables. For more explanation, see statistical analysis in this article [5,6].

**Figure 5 ijerph-20-03778-f005:**
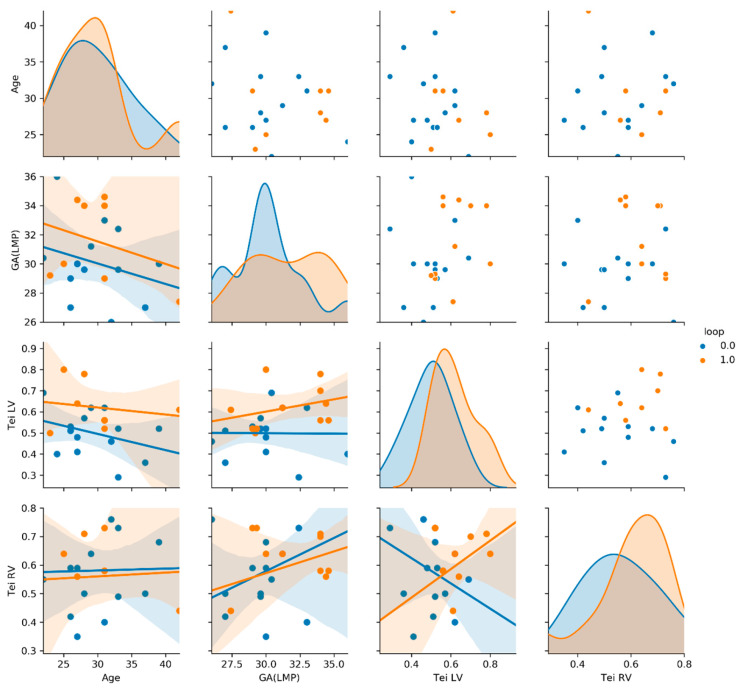
Correlogram of Tei LV, Tei RV, maternal age and gestational age in LGA with NC presented in red dots (loops 1,0) (*n* = 11) vs. LGA without NC = nNC group, presented in blue dots (loops 0,0) (*n* = 14) studied patients visualized with pair-plots in Python. A pair-plot allows one to observe both distributions of single variables and relationship between two variables. For more explanation see statistical analysis in this article [5,6].

**Table 1 ijerph-20-03778-t001:** Characteristics of groups of patients with: NC (loop 1) (*n* = 139) vs. nNC (loop 0) (*n* = 158), (t-Welch test). Characteristics of groups of patients with AGA (*n* = 263) and LGA (*n* = 25), (t-Welch test). The subgroup of fetuses with LGA: with NC (*n* = 14) and with no umbilical cord collision—nNC (*n* = 11) and significant statistical difference in TEI LV; *p* value ≤ 0.009 (t-Welch test).

	nNC *n* = 158	NC *n* = 139	*p* Value	Mean ± SD AGA *n* = 263	Mean ± SD LGA *n* = 25	*p* Value	LGA/nNC *n* = 14	LGA/NC *n* = 11	*p* Value
Tei RV (mean ± SD)	0.5 ± 0.2	0.5 ± 0.2	0.8	0.5 ± 0.2	0.6 ± 0.2	0.01	0.6 ± 0.2	0.6± 0.1	0.8
Tei LV (mean ± SD)	0.5 ± 0.2	0.5 ± 0.1	0.4	0.5 ± 0.2	0.6 ± 0.1	0.23	0.5 ± 0.1	0.6 ± 1	0.009
Maternal Age (mean ± SD)	30 ± 5.5	30 ± 5.3	0.9	30 ± 5.5	30 ± 5	0.73	30 ± 4.8	30 ± 5.8	0.94
GA(LMP) (mean ± SD)	30 ± 2.8	31 ± 3.3	10	30 ± 3	31 ± 2.7	0.94	30 ± 2.6	31 ± 2.7	0.2
AFI (mean ± SD)	16 ± 4.2	17 ± 4	0.4	16 ± 4	19 ± 2.8	0.0001	19 ± 2.8	20 ± 2.8	0.35

## Data Availability

The corresponding author keeps all data, which he can make available, if necessary. The datasets generated during and/or analyzed during the current study are available in the repository (juliamurlewska.jm@gmail.com).

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
