# Peer review of "The Presence of a Single Nuchal Cord in the Third Trimester May Not Affect Tei Index in LGA Fetuses"

_ijerph, 2023, doi:10.3390/ijerph20053778_

Round 1
Reviewer 1 Report
Dear Editor:
Thank you for inviting me to review the article:The presence of a single nuchal cord in the third trimester does not affect the right and left ventricular Tei index. This article mainly studies the effect of single umbilical cord around the neck on fetal cardiovascular function in the third trimester of pregnancy.
Major comments:
(1) The basis of the thesis is unsufficient, what kind of controversy or clinical problem is the scientific question based on, and the description is not sufficient and detailed.
(2) The U-shaped indentation of the umbilical cord should be checked with the true state of the umbilical cord around the neck in late delivery to rule out false positive diagnosis.
(3) Statistical methods are not concise. Please clarify the methods of difference analysis and correlation analysis.
(4) The basis of grouping is not scientific. Please explain the clinical basis of AGA, SGA and LGA grouping.
Minor comments:
(1) Please standardize the three - line table form.
Author Response
The presence of a single nuchal cord in the third trimester does not affect the right and left ventricular Tei index.
The title was changed into:
The presence of a single nuchal cord in the third trimester may not affect Tei index in LGA fetuses.
Reviewer 1
Major comments:
(1) The basis of the thesis is unsufficient, what kind of controversy or clinical problem is the scientific question based on, and the description is not sufficient and detailed. It was changed in the background
(2) The U-shaped indentation of the umbilical cord should be checked with the true state of the umbilical cord around the neck in late delivery to rule out false positive diagnosis.
The presence of the umbilical cord around the neck of the fetus was assessed in the third trimester of pregnancy. The course of the umbilical cord, as well as the position of the fetus, change during pregnancy, so it does not matter at all with the perinatal state, in our other work, which has not yet been published, and is currently awaiting review, we also assessed the tendency for the umbilical cord to persist until delivery.
The percentage of newborns wrapped with the umbilical cord during labour, who were also wrapped with the umbilical cord during echo-sonography in the third trimester of pregnancy was 37%, whereas in the case of foetuses who were not wrapped with the umbilical cord during the echo-sonographic examination but who were during labour was 18.6%.
(3) Statistical methods are not concise. Please clarify the methods of difference analysis and correlation analysis. I changed the tables with the data that I combined into one: table 1
(4) The basis of grouping is not scientific. Please explain the clinical basis of AGA, SGA and LGA grouping.
In methods it was changed: Large Gestational Age (LGA) was defined for ≥90th centiles and Small Gestational Age, (SGA) for ≤10th centiles
Minor comments:
- Please standardize the three - line table form- I changed the tables with the data that I combined into one: table 1
Reviewer 2 Report
The manuscript by Murlewska et al presents an analysis of the potential cardiac effects of fetal nuchal cord studied at the third trimester of gestational age. The authors measured RV and LV Tei indices for cardiac function in fetuses with different weight for gestational age.
Main results showed that the nuchal cord does not significantly affect fetal cardiac function regardless of weight for gestational age status, except in cases of LGA fetuses were LV Tei was higher versus LGA fetuses with no nuchal cord.
The manuscript is well written and conclusions are supproted by results. This dataset adds to the growing literature showing no significant effects of nuchal cord on fetal mortality and morbidity. In particular the focus on fetal cardiac function is appreciated.
The same group published similar results in 2021 (J Perinat Med. 2021 Feb 11;49(5):590-595.), where Tei index was determined in fetuses and the effects of nuchal cord was studied. This article should be cited and the differences with the manuscript in review should be clearly stated.
Reference list needs to be updated and corrected. Some references are abbreviated and some are not; for example “Pediatrics and Neonatology”, “Ultrasound in Obstetrics and Gynecology” (ref#48). Similarly, reference format needs to be standardized; for example in ref#47 Placenta, vol. 29, no. 11, pp. 976-981, 2008. Reference #12 was published in 2020, not in press anymore.
Author Response
Reviewer 2
The same group published similar results in 2021 (J Perinat Med. 2021 Feb 11;49(5):590-595.), where Tei index was determined in fetuses and the effects of nuchal cord was studied. This article should be cited and the differences with the manuscript in review should be clearly stated.- it was changed and now cited: In our previous publication, we also did not notice a significant impact of nuchal cord on fetal heart function expressed as Tei index, at the time of fetal heart examination (at mean gestational age 29+4 weeks), obtaining very similar Tei values for the left and right ventricles, as in this actual analysis [2]. In the discussion
Reference list needs to be updated and corrected. Some references are abbreviated and some are not; for example “Pediatrics and Neonatology”, “Ultrasound in Obstetrics and Gynecology” (ref#48). Similarly, reference format needs to be standardized; for example in ref#47 Placenta, vol. 29, no. 11, pp. 976-981, 2008. Reference #12 was published in 2020, not in press anymore.it was shortened and corrected, because some of the references had only a historical meaning, so it was actualized
Reviewer 3 Report
It is a very interesting study whose purpose was to evaluate the Tei index of the right ventricle and the left ventricle in fetuses with a circular umbilical cord around the neck through ultrasound. Demonstrating that umbilical cord circling to the neck does not contribute significantly to fetal cardiac dysfunction in the third trimester, regardless of fetal growth.
It is essential to mention the scope of the study, taking into account that it is intended to be published in a public health journal and that it can also be read by several pediatricians. In addition, it is important to mention the strengths and weaknesses of the study.
On lines 49 and 50 of the summary
You must mention the number of fetuses that corresponded to 45%, 48% and 100%
On lines 52 and 53 of the summary
It is necessary to verify these quantities 0.49 ± 0.1 against 0.61 ± 1; value of p=0.009, means that there are negative values.
On lines 61 and 63 of the background section
“When a nuchal cord is associated with compression, resistance to blood flow and fetal cardiac performance may be altered”. This statement must be justified with bibliographical references.
In the methods section
Prepregnancy mass index and gestational weight gain were considered as possible confounding variables. Since the association of these variables with the weight of the newborn has been demonstrated.
In the methods section, line 80
Whenever an abbreviation is used for the first time, it must be defined in the text (LMP).
In the methods section, line 82
The symbol must be less than and not greater than to define small for gestational age (SGA).
In the results section, line 127 to 128
Why was the evaluation not carried out at week 37 or 38, when it is considered that the fetus has already reached the expected size, and it is likely that the circulatory compromise is already evident?
In the results section line 130
AFI: Whenever an abbreviation is used for the first time in the text, it must be defined.
In the discussion section, line 206
Whenever an abbreviation is used for the first time in the manuscript, it must be defined; second, if the abbreviation is used only once, the term must be written in full and not use an abbreviation, the abbreviation is used when it is repeated more than 3 times the term you want to abbreviate
In the discussion section, line 214 to 215
If the degree of concordance is not previously evaluated, as well as the value of the precision, accuracy, and intra-observer variability of the ultrasounds performed, the fact that a single evaluator performs all the echocardiographies is not a strength, but rather a weakness.
What would be the scope of the study considering that it is intended to be published in a public health journal and can be read by several pediatricians?
It is suggested to delete the last paragraph of the conclusions and rather write a section of strengths and weaknesses of the study, where this paragraph can be included.
Author Response
Reviewer 3
Comments and Suggestions for Authors
It is essential to mention the scope of the study, taking into account that it is intended to be published in a public health journal and that it can also be read by several pediatricians. In addition, it is important to mention the strengths and weaknesses of the study.
The title was changed into:
The presence of a single nuchal cord in the third trimester may not affect Tei index in LGA fetuses.
And The basis of the thesis was changed in the background.
I added Strengths and weaknes of the study in the discussion:
Our publication paves the way for further research on the analysis of functional parameters of the heart of the fetus with NC depending on the occurrence of LGA or SGA. This needs to be studied in a larger population, although in the absence of foetal heart defects, the increased Tei index in foetuses with NC and growth impairment appears to be a temporary/transient effect and does not significantly affect the foetuses’ circulatory performance at the time of the study, as well as their postnatal circulatory performance.
And in the conclusions:
Single coil of umbilical cord around fetal neck in constitutional LGA fetuses may not be a cause of systolic and diastolic dysfunction of fetal heart, assessed by Tei index in this subgroup, however, fetuses with LGA may present higher Tei indices in general.
Therefore, we can assume that NC probably does not significantly affect the increase in Tei index in the group of fetuses with excessive growth in the third trimester of preg-nancy, but we need further study results on a larger group of patients to confirm these findings.
On lines 49 and 50 of the summary
You must mention the number of fetuses that corresponded to 45%, 48% and 100%
Umbilical cord around fetal neck occurred in 45% in AGA (NC in AGA n=118), in 48% in LGA (NC in LGA n=14) and 100% in SGA fetuses (NC in SGA n=9).
On lines 52 and 53 of the summary
It is necessary to verify these quantities 0.49 ± 0.1 against 0.61 ± 1; value of p=0.009, means that there are negative values.
I have corrected this in the table and in the text
On lines 61 and 63 of the background section
“When a nuchal cord is associated with compression, resistance to blood flow and fetal cardiac performance may be altered”. This statement must be justified with bibliographical references.
I have removed it from the text
In the methods section
Prepregnancy mass index and gestational weight gain were considered as possible confounding variables. Since the association of these variables with the weight of the newborn has been demonstrated.
I added a BMI > 30km/m2, as an exclusion from the study, which is true, because in first degree obesity, the quality of our ultrasound measurements decreases significantly and it is very difficult to carry out specific measurements, so we haven’t included such patients.
In the methods section, line 80
Whenever an abbreviation is used for the first time, it must be defined in the text (LMP).
I have changed it
In the methods section, line 82
The symbol must be less than and not greater than to define small for gestational age (SGA).
I changed it
In the results section, line 127 to 128
Why was the evaluation not carried out at week 37 or 38, when it is considered that the fetus has already reached the expected size, and it is likely that the circulatory compromise is already evident?
Our work was retrospective rather than prospective, whereas in 37/38 weeks of pregnancy, fetal heart echo is rarely performed.
In the results section line 130
AFI: Whenever an abbreviation is used for the first time in the text, it must be defined.
I have corrected the abbreviations
In the discussion section, line 206
Whenever an abbreviation is used for the first time in the manuscript, it must be defined; second, if the abbreviation is used only once, the term must be written in full and not use an abbreviation, the abbreviation is used when it is repeated more than 3 times the term you want to abbreviate
I have corrected it
In the discussion section, line 214 to 215
If the degree of concordance is not previously evaluated, as well as the value of the precision, accuracy, and intra-observer variability of the ultrasounds performed, the fact that a single evaluator performs all the echocardiographies is not a strength, but rather a weakness.
I have changed it in the text (discussion).
What would be the scope of the study considering that it is intended to be published in a public health journal and can be read by several pediatricians?
It is suggested to delete the last paragraph of the conclusions and rather write a section of strengths and weaknesses of the study, where this paragraph can be included.
The title was changed into:
The presence of a single nuchal cord in the third trimester may not affect Tei index in LGA fetuses.
And The basis of the thesis was changed in the background.
I added Strengths and weaknes of the study in the discussion:
Our publication paves the way for further research on the analysis of functional parameters of the heart of the fetus with NC depending on the occurrence of LGA or SGA. This needs to be studied in a larger population, although in the absence of foetal heart defects, the increased Tei index in foetuses with NC and growth impairment appears to be a temporary/transient effect and does not significantly affect the foetuses’ circulatory performance at the time of the study, as well as their postnatal circulatory performance.
And in the conclusions:
Single coil of umbilical cord around fetal neck in constitutional LGA fetuses may not be a cause of systolic and diastolic dysfunction of fetal heart, assessed by Tei index in this subgroup, however, fetuses with LGA may present higher Tei indices in general.
Therefore, we can assume that NC probably does not significantly affect the increase in Tei index in the group of fetuses with excessive growth in the third trimester of preg-nancy, but we need further study results on a larger group of patients to confirm these findings.